# Immune signaling of *Litopenaeus vannamei* c-type lysozyme and its role during microsporidian *Enterocytozoon hepatopenaei* (EHP) infection

**Nutthapon Sangklai**[1], **Premruethai Supungul**[2], **Pattana Jaroenlak**[1]*,
**Anchalee Tassanakajon**[1]*

**1** Center of Excellence for Molecular Biology and Genomics of Shrimp, Department of Biochemistry, Faculty of Science, Chulalongkorn University, Bangkok, Thailand, **2** Aquatic Molecular Genetics and Biotechnology Research Team, National Center for Genetic Engineering and Biotechnology (BIOTEC), National Science and Technology Development Agency (NSTDA), Pathumthani, Thailand

\* pattana.j@chula.ac.th (PJ); anchalee.k@chula.ac.th (AT)

**Data Availability Statement:** All data generated or analyzed during this study are included in the

## Abstract

The microsporidian *Enterocytozoon hepatopenaei* (EHP) is a fungi-related, spore-forming parasite. EHP infection causes growth retardation and size variation in shrimp, resulting in severe economic losses. Studies on shrimp immune response have shown that several anti-microbial peptides (AMPs) were upregulated upon EHP infection. Among those highly upregulated AMPs is c-type lysozyme (*Lv*Lyz-c). However, the immune signaling pathway responsible for *Lv*Lyz-c production in shrimp as well as its function against the EHP infection are still poorly understood. Here, we characterized major shrimp immune signaling pathways and found that Toll and JAK/STAT pathways were up-regulated upon EHP infection. Knocking down of a *Domeless* (*DOME*) receptor in the JAK/STAT pathways resulted in a significant reduction of the *Lv*Lyz-c and the elevation of EHP copy number. We further elucidated the function of *Lv*Lyz-c by heterologously expressing a recombinant *Lv*Lyz-c (r*Lv*Lyz-c) in an *Escherichia coli*. r*Lv*Lyz-c exhibited antibacterial activity against several bacteria such as *Bacillus subtilis* and *Vibrio parahaemolyticus*. Interestingly, we found an antifungal activity of r*Lv*Lyz-c against *Candida albican*, which led us to further investigate the effects of r*Lv*Lyz-c on EHP spores. Incubation of the EHP spores with r*Lv*Lyz-c followed by a chitin staining showed that the signals were dramatically decreased in a dose-dependent manner, suggesting that r*Lv*Lyz-c possibly digest a chitin coat on the EHP spores. Transmission electron microscopy analysis revealed that an endospore layer, which is composed mainly of chitin, was digested by r*Lv*Lyz-c. Lastly, we observed that EHP spores that were treated with r*Lv*Lyz-c showed a significant reduction of the spore germination rate. We hypothesize that thinning of the endospore of EHP would result in altered permeability, hence affecting spore germination. This work provides insights into shrimp immune signaling pathways responsible for *Lv*Lyz-c production and its anti-EHP property. This knowledge will serve as important foundations for developing EHP control strategies.

published manuscript and its supplementary information files.

**Funding:** NS received a scholarship from the Second Century Fund (C2F), Chulalongkorn University and the 90th Anniversary of Chulalongkorn University Fund (Ratchadaphiseksomphot Endowment Fund). AT would like to thanks Thailand Science Research and Innovation Fund Chulalongkorn University, and Chulalongkorn University under Ratchadaphisek Somphot Endowment Fund to the Center of Excellence for Molecular Biology and Genomics of Shrimp (CEMS). PJ would like to acknowledge the Office of the Permanent Secretary, Ministry of Higher Education, Science, Research and Innovation (OPS MHESI), Thailand Science Research and Innovation (TSRI), and Chulalongkorn University for a research grant for new scholars (Grant No. RGNS 65-013) and a grant for development of new faculty staff, Ratchadaphiseksomphot Fund (Grant No. DNS 65_033_23_003_1). The funders had no role in study design, data collection and analysis, decision to publish, or preparation of the manuscript.

**Competing interests:** The authors have declared that no competing interests exit.

## Author summary

The microsporidian parasite *Enterozytozoon hepatopanei* (EHP) has caused a significant disease in Penaeid shrimp since 2009. EHP infection results in growth retardation and severe size variation in shrimp. Currently, the immune defense mechanism against EHP infection is still largely unknown, thus the basic knowledge on shrimp innate immunity is useful for controlling EHP infection. Recently, it has been shown that a lysozyme-c type (*Lv*Lyz-c) potentially plays an important role during the EHP infection. However, the signaling cascade that regulates the *Lv*Lyz-c production is poorly studied. Here, we examined the immune signaling pathway responsible for the *Lv*Lyz-c production and its crucial role in controlling EHP infection. We showed that Toll and JAK/STAT pathways were up-regulated upon EHP infection. Inhibition of JAK/STAT pathway resulted in reduction of *Lv*Lyz-c and increase in EHP proliferation. Further characterization using a recombinant *Lv*Lyz-c (r*Lv*Lyz-c) revealed both antibacterial and antifungal activities. r*Lv*Lyz-c digested an endospore layer of the EHP spore. Thinning of the endospore consequently reduced spore germination rate. This work provides a basic foundation on the immune signaling pathway leading to the *Lv*Lyz-c production and its function against EHP spores. *Lv*Lyz-c could serve as a promising target for the development of EHP control strategy.

## Introduction

Shrimp farming industry has a high economic value in many countries around the world [1]. The high demand in international markets and global consumption result in a rapid expansion of the shrimp aquaculture industry, especially a culture of the Pacific white shrimp *Litopenaeus vannamei*, which can grow faster and can be cultured at a high density [2]. However, the major obstacle in successful shrimp farming comes from several disease outbreaks caused by bacterial, viral, and parasitic infections [3–5]. These infections lead to significant economic losses [6]. Recently, an emerging disease called hepatopancreatic microsporidiosis (HPM), has become a major concern in the shrimp farming industry [7,8]. HPM is caused by a microsporidian parasite *Enterocytozoon hepatopenaei* (EHP) [9]. EHP infection is associated with growth retardation and severe size variation, resulting in a reduction in shrimp biomass production [10].

EHP was first reported to infect the black tiger shrimp *Penaeus monodon* in Thailand in 2004 [11]. Since then, EHP has been widespread in many Asian countries, for example, Korea, China, Indonesia, India, Vietnam, and Malaysia [5,12–16]. Although the EHP infection does not cause mortality in shrimp, the co-infection of EHP with bacteria and viruses including *Vibrio parahaemolyticus* caused acute hepatopancreatic necrosis disease (VP$_{AHPND}$), white spot syndrome virus (WSSV), and myonecrosis virus (IMNV) can cause a 100% mortality [17–19]. It is suggested that EHP infection could make the shrimp weaken. Hence, they are more susceptible to other diseases [20].

Shrimp immunity against pathogens heavily relies on innate immunity, including cellular and humoral responses [21]. In cellular immune response, hemocytes play an important role in protection against pathogen invasions [22]. The cellular immune responses performed directly by hemocytes are apoptosis, nodulation, encapsulation, and phagocytosis [23]. While, the humoral immune responses include prophenoloxidase (proPO) system, blood clotting system, and antimicrobial peptide (AMPs) [24]. Both cellular and humoral immune responses work together to limit the invading pathogens [25]. The defense against these pathogens is

specifically mediated by pattern recognition receptors (PRRs), which bind to pathogen-associated molecular patterns (PAMPs) [26]. The recognition of PAMPs induces several immune signaling cascades, including Toll like receptor (TLR), immune deficiency (IMD), and JAK/STAT signaling pathways [27–29]. The activation of the signaling cascade triggers a secretion of antimicrobial peptides (AMPs), such as penaeidins, lysozymes, crustins, and anti-lipopolysaccharide factor (ALF) to fight against bacterial and viral infection [21].

In humans, microsporidia are recognized by macrophages via MyD88-dependent TLR2 and stimulate the expression of several cytokines and chemokines [30]. In addition, MyD88 signaling is required for the activation of dendritic cells by *Enterocytozoon bieneusi*, suggesting a potential function in activation of downstream molecules. This results in the production of cytokine and AMPs, which plays an important role in limiting microsporidia infection [31]. In silkworm *Bombyx mori*, the Toll, IMD, and JAK/STAT signaling pathways are induced by microsporidian *Nosema bombycis* infection, which leads to the production of antimicrobial peptides including lebocin, gloverin, cecropin, and attacin families [32,33]. Similar to silkworm, these signaling pathways in shrimp are important in EHP clearance by increasing proPO activating cascade and several immune-related genes to combat EHP proliferation [34].

Immune-related factors in Toll, IMD, and JAK/STAT signaling pathways were involved in the cellular defense mechanism against microsporidian infection [35]. In shrimp, some AMPs have shown to exhibit antifungal activity [36,37] implying their potential role in EHP infection. Previous study revealed that α-2 macroglobulin, c-type lectin, peritrophin-44-like protein, lysozyme-c type, prophenoloxidase activating enzyme, and integrin were up-regulated during the EHP infection [38]. Of interest, the proteomic and transcriptomic studies showed that the expression level of lysozyme-c type from *L. vannamei* (*Lv*Lyz-c) was approximately increased by 3.1 and 1.5-fold after the EHP infection, respectively [38,39]. This indicates that *Lv*Lyz-c plays an essential role in response to the EHP infection. However, the information on innate immunity response during EHP infection, especially the signaling pathway controlling the lysozyme production and the function of lysozyme against the EHP are still unclear. Here, we investigated the innate immune signaling pathways responsible for the *Lv*Lyz-c production upon EHP infection using an RNA interference technique. To characterize the role of the *Lv*Lyz-c, recombinant *Lv*Lyz-c (r*Lv*Lyz-c) was produced in the *Escherichia coli* system. r*Lv*Lyz-c was tested for its antimicrobial activity and ability to digest chitin layer of EHP endospore and inhibit spore germination. This work provides insights into the molecular mechanism on how *Lv*Lyz-c is regulated and the important function of *Lv*Lyz-c in limiting EHP infection. *Lv*Lyz-c could serve as one of the promising targets for controlling EHP infection problems in the future.

## Results

### Several immune-related genes are up-regulated during EHP infection

Molecular mechanisms on how shrimp respond to EHP infection remain largely unknown. Here, we systematically quantify the expression levels of several immune related-genes in the JAK/STAT, TLR pathways, and AMPs upon the EHP infection. For the JAK/STAT pathway, a receptor *LvDOME* was significantly up-regulated at 9 and 15 days post-cohabitation (Fig 1A), while *LvJAK* was up-regulated at day 1 and 9 (Fig 1B). In contrast, a transcription factor *LvSTAT* was significantly increased only at day 9 (Fig 1C). In the TLR pathway, the *LvTLR2* receptor was increased only at the 11 day post-cohabitation (Fig 1D). However, the downstream effector *LvMyD88* was up-regulated at all timepoints (Fig 1E), while *LvDorsal* was up-regulated at day 1 and 15 (Fig 1F). Our results suggest that both JAK/STAT and TLR pathways are involved in the EHP infection. However, it is unclear which immune signaling pathway

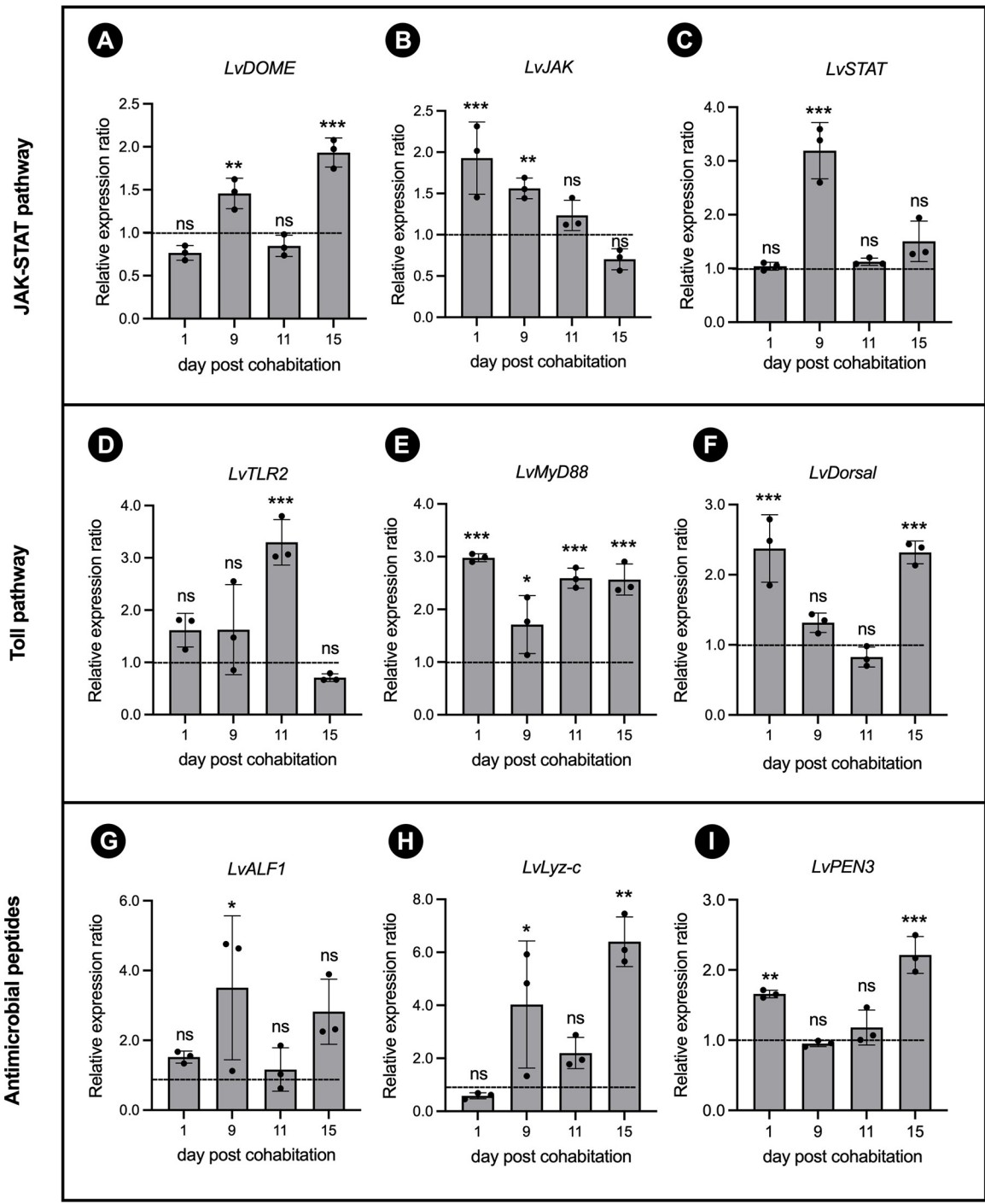

**Fig 1. Temporal expression of the immune-related genes upon EHP infection.** (A-C), JAK/STAT pathway; (D-F), Toll like receptor (TLR) pathway; (G-I), Antimicrobial peptides (AMPs). Shrimp were reared together with EHP-positive shrimps. Immune-related gene transcriptional levels were determined via qRT-PCR at 1, 9, 11, and 15 days after EHP cohabitation. Each data point was normalized to the expression of the EF-1α gene and calculated relative to the expression level in the specific pathogen-free (SPF) shrimp. Dotted line represents the expression level in the SPF shrimp. Error bars represent the standard deviation of three biological replicates (n = 3). All the data were analyzed by one-way ANOVA. * represents $P<0.05$, ** shows $P<0.01$, and *** indicates $P<0.001$.

may play a role in an early or late stage of the infection. In response to the EHP infection, we found that *LvALF1* was significantly up-regulated at 9 days post-cohabitation (Fig 1G), while *LvPEN3* was increased at day 1 and 15 (Fig 1I). Interestingly, the expression level of *LvLyz-c* was increased by approximately 6.4 folds at day 15, which was higher than other AMPs (Fig 1H). This highlighted that *LvLyz-c* possibly be one of the important AMPs or alternatively working together with other AMPs to fight against the EHP infection.

### *LvTLR2* and *LvDOME* are required for suppressing EHP proliferation in shrimp

To test whether *LvLyz-c* is responsible for limiting EHP infection, we silenced 2 main surface receptors, including *LvTLR2* and *LvDOME* using an RNA interference (RNAi) technique. Double-stranded RNA (dsRNA) specific to either *LvTLR2* or *LvDOME* were injected twice prior to cohabitation (See method). Our results showed that *LvTLR2* and *LvDOME* were successfully suppressed and the silencing effect lasted until 11 days post-cohabitation (Fig 2A and 2D). After knocking down these two receptors, we tested the *LvLyz-c* expression level at different timepoints (Fig 2B and 2E). *LvDOME* knockdown resulted in reduction of the *LvLyz-c* level at 7, 9, and 11 days post cohabitation (Fig 2B), suggesting that the production of *LvLyz-c* could be under the regulation of *LvDOME*. To further characterize the effect of *LvLyz-c* on the EHP proliferation, we quantified the EHP copy number in the shrimp that were injected with ds*LvDOME* compared to the control group. When *LvDOME* was knocked down, EHP copy number significantly increased at least 10 times (Fig 2C). This result supported the functions of JAK/STAT pathways in reducing EHP infection.

In contrast, *LvTLR2* knockdown increased the *LvLyz-c* expression level (Fig 2E). This suggests that *LvLyz-c* may not be regulated by the TLR pathway. However, *LvTLR2* knockdown could result in increasing the EHP copy number at 9 and 11 days post-cohabitation (Fig 2F), indicating that the TLR pathway is still one of the immune signaling pathways responsible for controlling the EHP infection in shrimp. However, further investigation needs to be performed to elucidate the function of *LvTLR2* upon EHP infection.

### *Lv*STAT potentially binds to the *LvLyz-c* promoter

To further confirm that the *LvLyz-c* production is under the JAK/STAT signaling pathways, we harnessed the luciferase reporter assay. Briefly, the transcription factor *LvSTAT* coding sequence was cloned into a pcDNA3.1 vector. The predicted promoter sequence of *LvLyz-c* (positions −213 to +24, 238 bp) was cloned into plasmid pGL3 containing a luciferase reporter gene (*luc*+) (Fig 3A). After co-transfection of these two plasmids into HEK293 cells, we found that the relative luciferase activity significantly increased by 2.88 folds ($P<0.0001$) compared to the control groups, including no plasmid, pGL3, and pcDNA3.1 transfected cells. Meanwhile, the signal disappeared in cells carrying a deletion of the *STAT* binding site (TTCTCA-GAAA [40], 10 bp) construct (Fig 3B). This result confirms that *Lv*STAT binds to the *LvLyz-c* promoter and the *LvLyz-c* is under the regulation of the JAK/STAT signaling pathway.

### r*Lv*Lyz-c exhibits both antibacterial and antifungal activities

Previous study has shown that lysozyme displayed antimicrobial activities against various pathogens [41]. To test whether *Lv*Lyz-c from shrimp contains any antimicrobial activities, we recombinantly expressed *Lv*Lyz-c in an *E. coli* system. The recombinant *Lv*Lyz-c (r*Lv*Lyz-c) was successfully purified using a Ni-NTA affinity chromatography (S1 Fig) and the agar well diffusion assay was performed against gram-positive bacteria, gram-negative bacteria, and yeast cells (Fig 4A). Clear zones were observed only when tested with a gram-positive *B*.

 

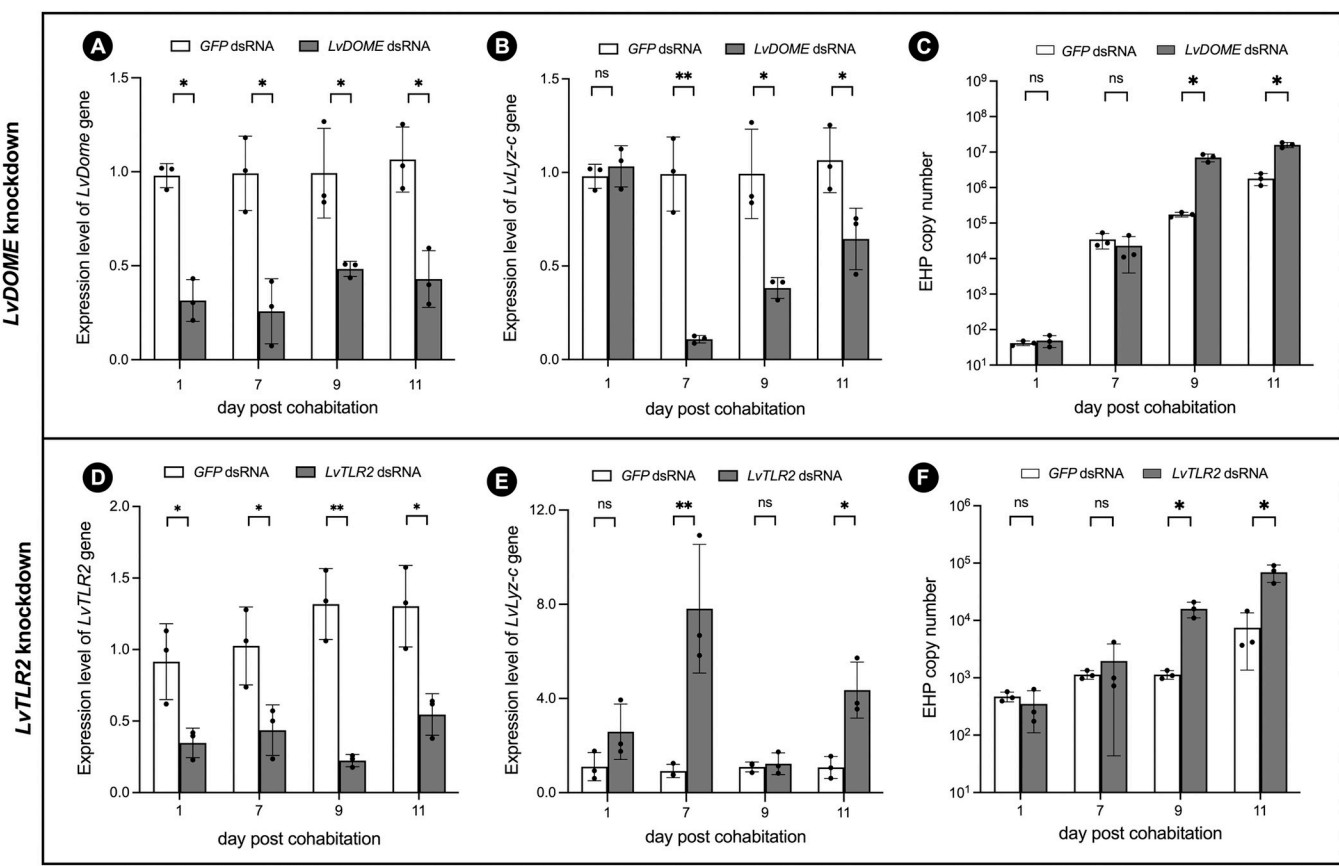

**Fig 2. Effect of the immune-related receptor knockdown on *LvLyz-c* expression level and EHP copy number.** (A-C) Domeless receptor and (D-F) Toll-like receptor 2 (TLR2) knockdowns. The mRNA expression level of *LvDOME*, *LvTLR2*, and *LvLyz-c* were quantified by qRT-PCR. EHP copy numbers were analyzed using absolute qPCR comparing with a*SSU* rRNA gene of EHP. The GFP dsRNA treated shrimp were used as a control. All the data were analyzed by one-way ANOVA. bars with * indicate statistically significant differences (P<0.05); bars with ** indicate highly statistically significant (P<0.01).

*subtilis*, and two of the gram-negative bacteria *V. parahaemolyticus* (AHPND stain) and *V. haveyi* (Fig 4A and 4B). It is unclear why r*Lv*Lyz-c selectively works on some bacterial species but not others. It is possible that different bacteria process different modifications on the cell wall layers, which could affect the *Lv*Lyz-c activity [42]. Interestingly, r*Lv*Lyz-c exhibited the antifungal activity against *C. albican* (Fig 4). This implies that r*Lv*Lyz-c could potentially digest both peptidoglycan and chitin, a major component of bacteria and fungi cell walls. This result opens a possibility that r*Lv*Lyz-c might be able to digest a chitin layer on the EHP spore.

## r*Lv*Lyz-c digests a chitin layer presented on the EHP spore

Typically, microsporidian spores contain 2 different layers, including a proteinaceous electron-dense exospore and electron-lucent chitinous endospore [43]. To investigate the role of *Lv*Lyz-c against the EHP spores, we stained the mature EHP spores with a chitin staining dye, called calcofluor white M2R. After incubating the spores with various concentrations of r*Lv*Lyz-c ranging from 0.1 μM to 16 μM, the fluorescent intensities were significantly reduced in a dose-dependent manner (Fig 5A and 5B). The intensity was the lowest at 8 μM concentration and remained unchanged when the concentration was increased to 16 μM (Fig 5B). To further investigate the effect of r*Lv*Lyz-c on the EHP spore layers in a nanoscale resolution, we utilized a room-temperature transmission electron microscopy (TEM). We measured the area

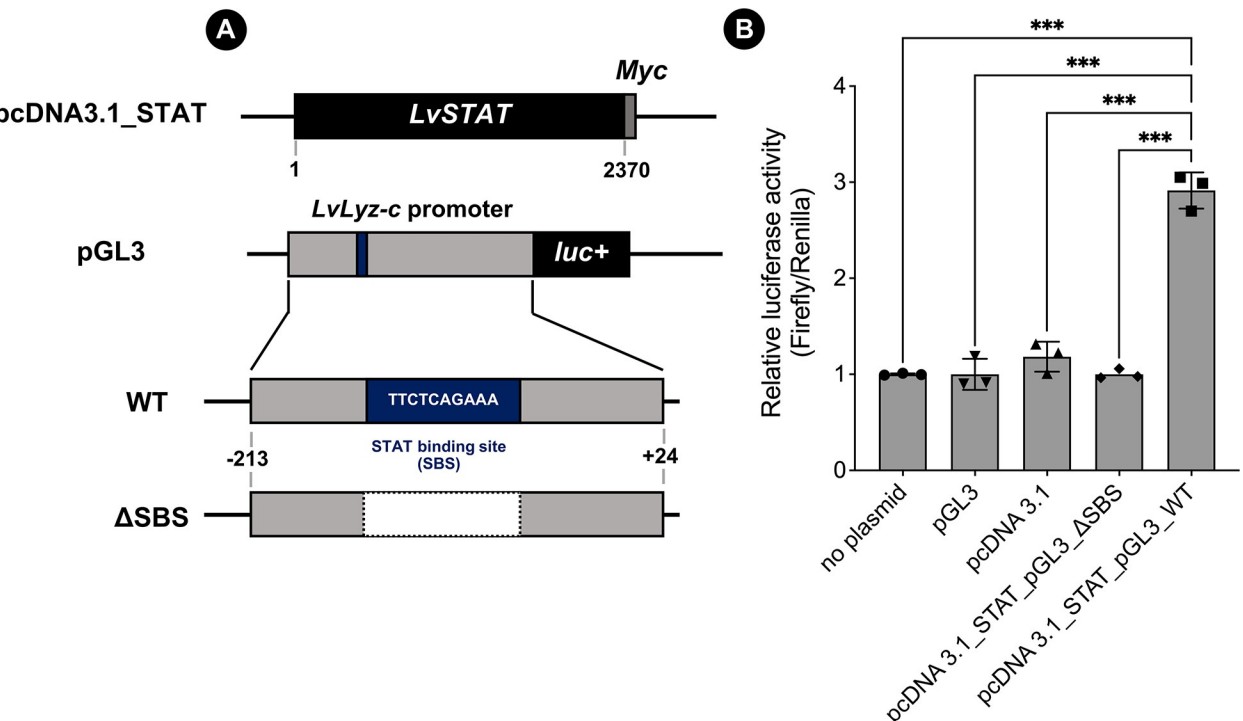

**Fig 3. Luciferase assay of *LvSTAT* and *LvLyz-c* promoter transiently expressed in the HEK293 cells.** (A) Schematics of a luciferase reporter gene and constructs of protein expression plasmids. ΔSBS represents a construct where the *STAT* binding site (10 bp) was removed. (B) Transient luciferase assay detection. Values are means ± standard errors of three independent replicates ($n = 3$). Statistical significance was determined by student T-test (** $P < 0.01$).

and the thickness of both spore wall layers. The results showed that endospore thickness and area significantly decreased after being treated with 16 μM of rLvLyz-c (Fig 5C–5E). However, the exosore layer remains unaffected (Fig 5D and 5E). These results suggest that rLvLyz-c exhibits a chitinase activity and directly digests the EHP endospore layer.

### Recombinant *Lv*Lyz-c reduces EHP spore germination

Next, we further investigate the role of rLvLyz-c on EHP spore germination. A recent finding on mechanisms of the polar tube firing in microsporidia showed that the spore wall is one of the essential components to provide a successful spore germination and transport of infectious cargo [44]. Hence, we hypothesized that thinning of the endospore layer by rLvLyz-c could affect the EHP spore germination. To test this hypothesis, mature EHP spores were incubated with different concentrations of rLvLyz-c varying from 0.3125 μM to 5 μM and the spore germination was enduced by 2% Phloxine B solution. Our results revealed that the germination rate drastically decreased from ~60% to 7% after incubating with 5 μM rLvLyz-c (Fig 6A and 6B). Collectively, our results suggest that rLvLyz-c reduces the thickness of the endospore layer and this contributes to the reduction of spore germination.

### Discussion

Toll and JAK/STAT signaling pathways participate in the synthesis of AMPs against various pathogens [45]. Synthesizing from our data, we propose a model of how *Lv*Lyz-c is regulated and how *Lv*Lyz-c functions in limiting EHP infection in shrimp (Fig 7). Our results show that the JAK/STAT pathway mediates the *LvLyz-c* production in response to EHP infection and the

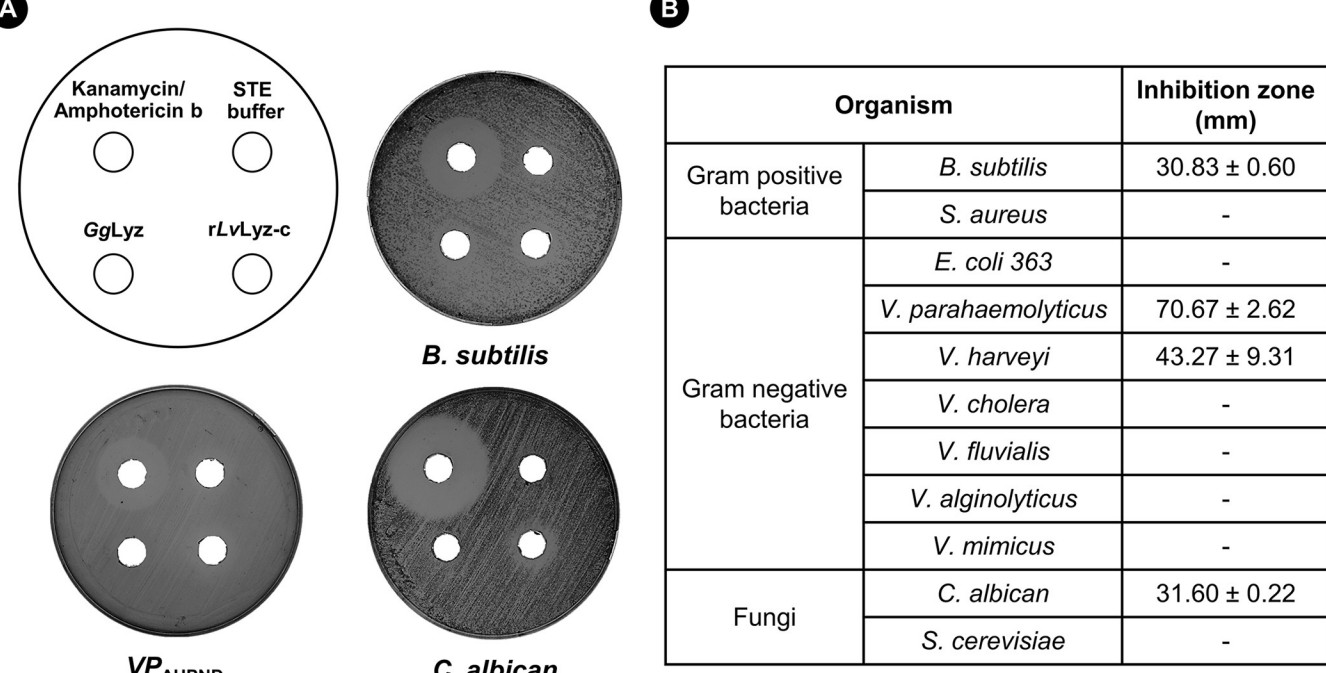

**Fig 4. Agar diffusion antimicrobial assay of a recombinant *Lv*Lyz-c.** (A) Antimicrobial plate assay against gram negative bacteria *Vibrio parahaemolyticus*, gram positive bacteria *Bacillus subtilis*, and yeast *Candida albican*. (B) Clear zone inhibition measurement shown in millimeter units. Values are means ± standard errors of three independent replicates (*n* = 3).

production of *Lv*Lyz-c is independent of the Toll signaling pathway. A transcription factor STAT binds to a *LvLyz-c* promoter which leads to the production of *Lv*Lyz-c. Functional characterization of *Lv*Lyz-c reveals the anti-EHP properties by (1) digesting a chitinous endospore layer of EHP spore and (2) inhibiting the EHP spore germination process (Fig 7). However, the mechanistic details of how *Lv*Lyz-c could prevent the spore germination remain an open question and require further investigation.

The JAK/STAT signaling pathway is an essential pathway associated with both innate and adaptive immunity [22,23]. In vertebrates, various cytokines such as interferons and interleukins activate the JAK/STAT pathway, resulting in mediated immune responses to infections [24]. In silkworms, the JAK/STAT signaling pathway is shown to be associated with antifungal immune response [46]. The inhibition of the JAK/STAT pathway could significantly decrease the antifungal activity of the hemolymph against *Beauveria bassiana* infection [25]. In shrimps, the JAK/STAT signaling pathway plays an important role against bacteria and virus infection [47,48]. A recent study shows that the expression levels of *PmDOME* and *PmSTAT*, were significantly increased after a white spot syndrome virus (WSSV) infection [49]. Disruption of *PmDOME* or *PmSTAT* by RNA interference affected the prophenoloxidase system and the production of IFN-like antiviral cytokine and AMPs during the WSSV infection [49,50]. A previous study has also suggested that the production of AMPs is controlled by the JAK/STAT signaling pathways [29,47]. Here, we demonstrate that *LvDOME* is one of the receptors in response to the EHP infection. The signaling cascade is activated through the JAK/STAT signaling pathway to stimulate the expression of *Lv*Lyz-c, to fight against the EHP infection in shrimp.

In invertebrates, lysozyme is one of the key components in innate immunity. Its function is involved in hydrolyzing bacterial cell walls [27]. Previous studies have demonstrated that lysozymes from *P. monodon* exhibited antimicrobial activities against both gram-positive and

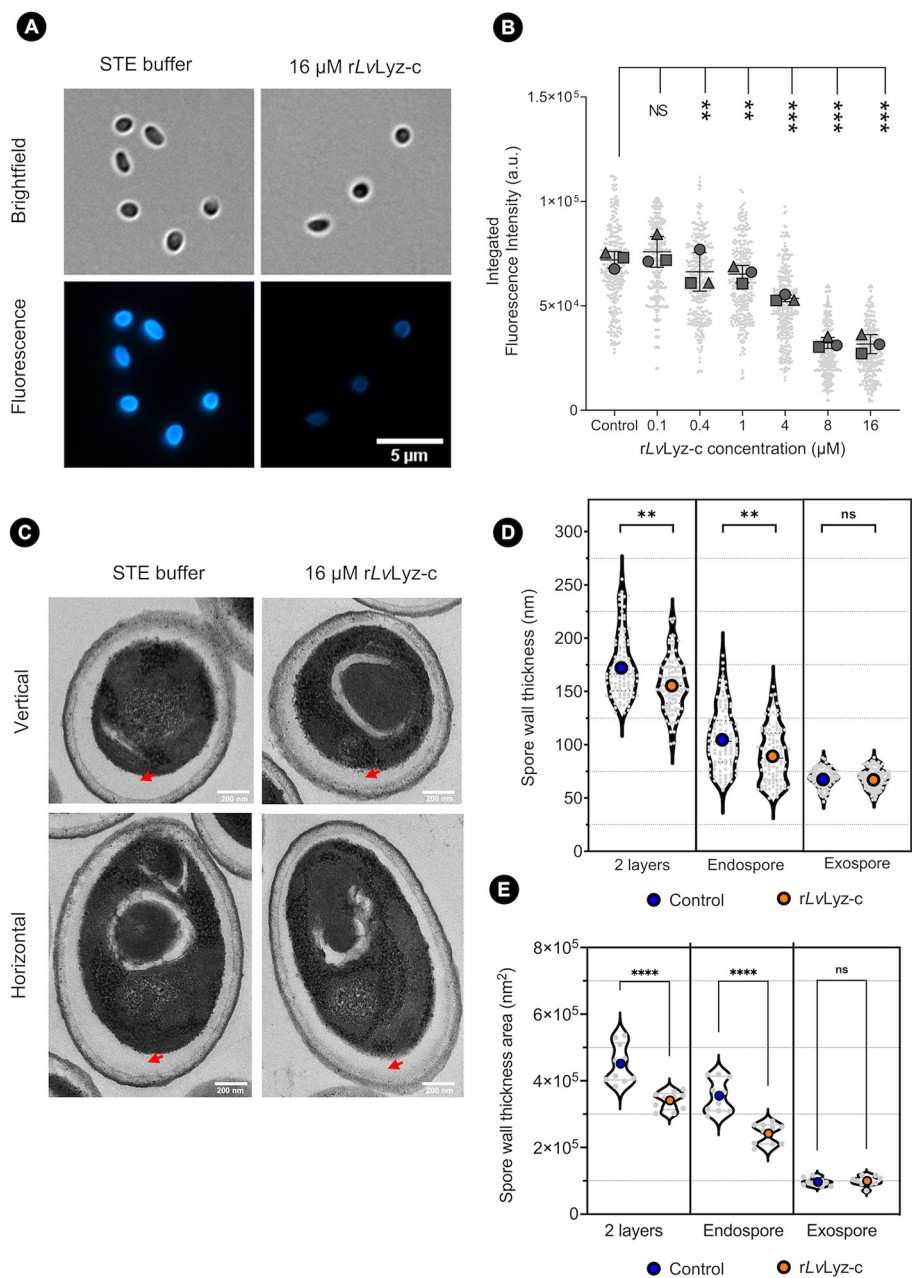

**Fig 5. Enzymatic digestion of the EHP endospore layer.** (A) Fluorescent micrographs of EHP spores stained with a chitin dye. (B) Quantification of the fluorescence intensity of EHP spores from (A). (C) Representative TEM micrographs. Red arrows represent the endospore layer. (D) Quantification of the EHP spore wall thickness and (E) spore wall area. Each experiment was performed in three biological replicates (n = 100 for each replicate, except in (E) that n = 10). ** $P<0.01$ and *** ($P<0.001$). Scale bars for the micrographs are 5 μm for fluorescence analysis and 200 μm for TEM.

gram-negative bacteria [28,29]. Meanwhile, proteomic and transcriptomic studies showed that the expression level of *Lv*Lyz-c was increased during the EHP infection, indicating that *Lv*Lyz-c might play an important role in response to the EHP infection [38,39]. In accordance with the previous work [51], r*Lv*Lyz-c showed similar antibacterial activities. Unexpectedly, we found the antifungal activity of r*Lv*Lyz-c against *C. albican* and EHP spores. These two

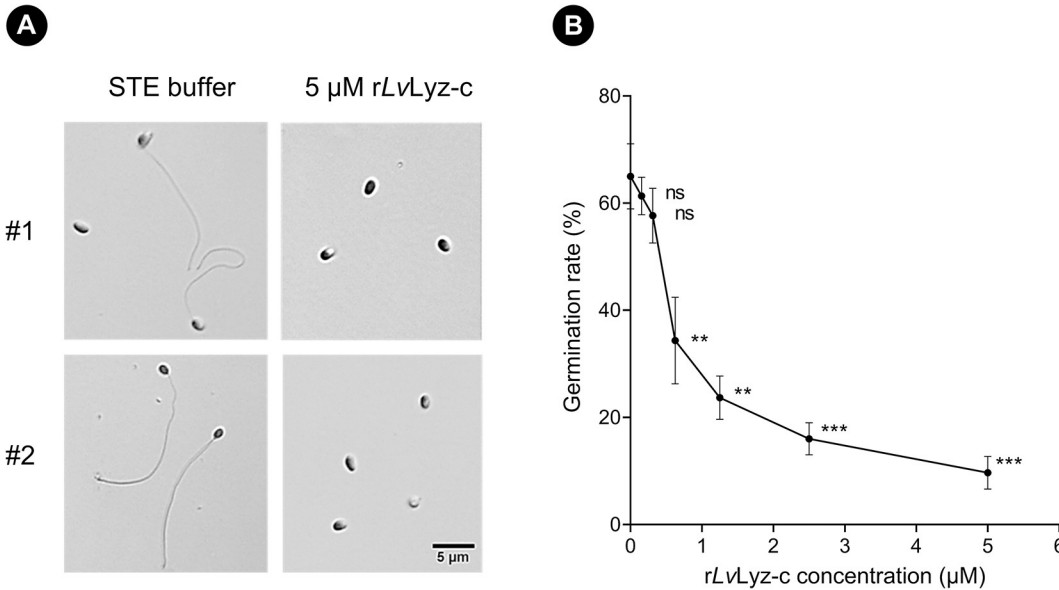

**Fig 6. Inhibitory effect of rLvLyz-c on EHP spore germination.** (A) EHP spore germination after incubating with rLvLyz-c (B) Quantification of the EHP spore germination rate. Each experiment was performed in three independent biological replicates (n = 100 for each replicate). ** $P<0.01$ and *** $P<0.001$. Scale bar is 5 μm.

organisms shared a similar cell wall architecture, which is composed of chitin. Hence, it is possible that rLvLyz-c could digest the chitin layer. Indeed, our data shows that the thickness of the EHP endospore layer–a layer riched in chitin, is reduced after the rLvLyz-c treatment. This suggests its role in limiting the EHP infection. In addition to the chitinase property, rLvLyz-c possibly alters the membrane permeability of the spore which affects the ion flux and osmotic pressure required for successful spore germination. A recent work on a biophysical characterization of the microsporidian germination process suggests that the spore wall is important to withstand the pressure generated during the germination process [44]. Hence, the reduction of the EHP endospore layer by rLvLyz-c might disturb an influx of water into the spore, resulting in an osmotic pressure imbalance which hinders spore germination.

Taken together, our data demonstrate that the JAK/STAT signaling pathway is responsible for the production of the LvLyz-c in response to the EHP infection. LvLyz-c exhibits antifungal activity that limits the EHP germination process. This study provides insights into the defense mechanism of shrimp against the EHP infection and the roles of LvLyz-c in limiting the EHP germination process. The LvLyz-c could serve as a promising target to be used as one of the EHP prevention and control strategies.

## Materials and methods

### Ethics statement

This study was carried out in strict accordance with the recommendations of the Weatherall report. The protocol was approved by the Committee on the Ethics of Animal Experiments of Chulalongkorn University (Protocol Number: 2323018).

### Bacterial strains and animals

For the antimicrobial assays, Gram-positive bacteria (*Bacillis subtilis* and *Staphylococcus aureus*), Gram-negative bacteria (*Vibrio harveyi*, *Vibrio parahaemolyticus*, *Vibrio fluvialis*,

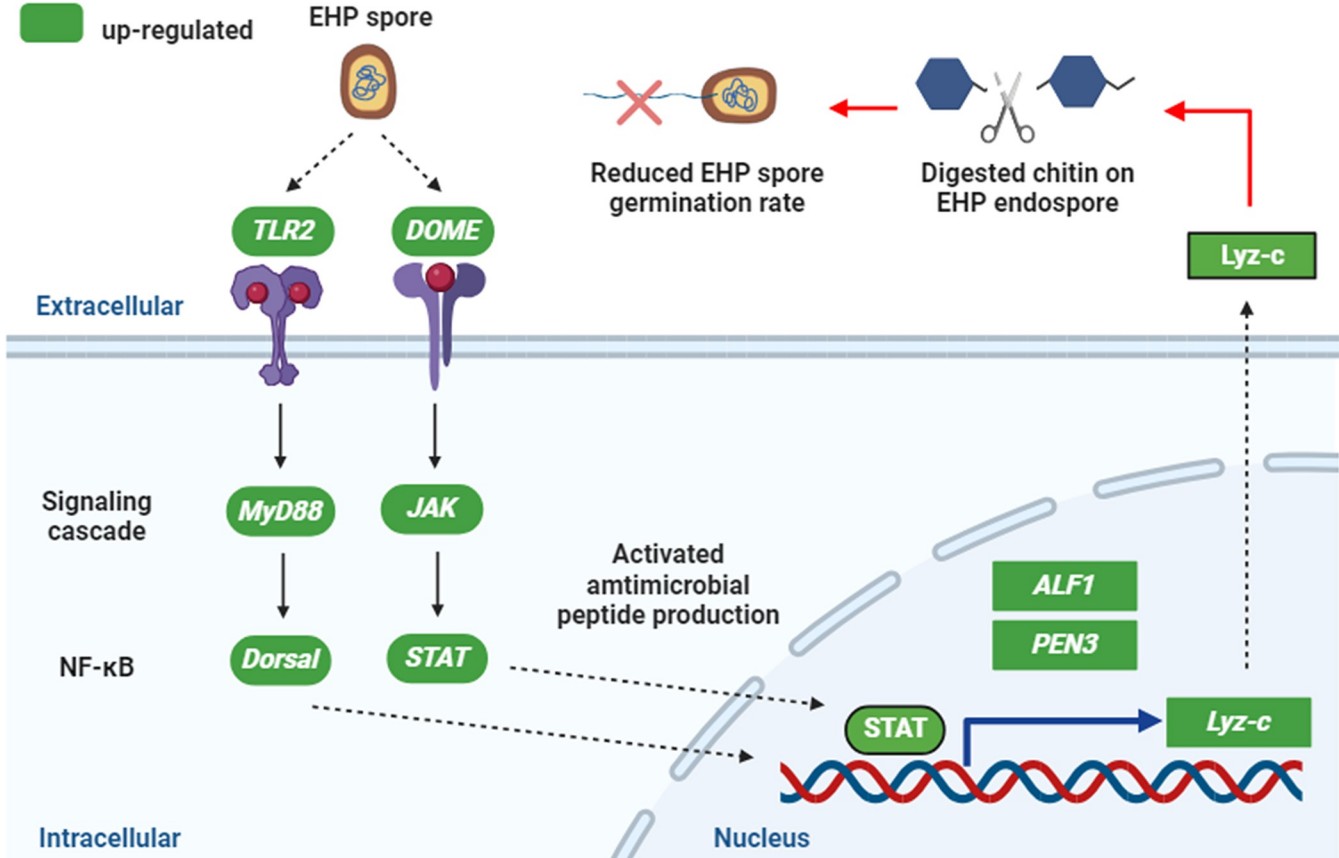

**Fig 7. Schematic representation of EHP-mediated activation of the Lyz-c production by TLR2 or JAK-STAT signaling pathway.** EHP induces TLR2 and DOME receptors to activate their downstream effectors, *LvMyD88* and *LvJAK*, respectively. *LvDorsal* or *LvSTAT* transcription factor promotes a specific set of antimicrobial peptides, including *LvLyz-c*, *LvPEN3*, and *LvALF1*. *Lv*Lyz-c production upon the infection could digest the endospore layer of EHP and inhibit the EHP germination process. Note that Fig 7 was created with BioRender.

*Vibrio alginolyticus*, *Vibrio cholera*, *Vibrio mimicus*, and *Escherichia coli* 363), and yeast cells (*Candida albicans*) were used. Furthermore, *E. coli* Rosetta(DE3) pLysS was used for the recombinant protein expression experiment.

Specific pathogen-free (SPF) juvenile *Litopenaeus vannamei* with an average weight between 2–4 g were provided by the Marine Shrimp Broodstock Research Center II (MSBRC-2), Charoen Pokphand Foods PCL (Phetchaburi Province, Thailand). Shrimps were acclimated under laboratory conditions at an ambient temperature of 28°C ± 1°C and with 20 ppt salinity for 1 week prior to experiments.

## EHP spore preparation

EHP-infected shrimps were obtained from commercial ponds in Chanthaburi province, Eastern Thailand. Hepatopancreas from 15–20 shrimps were pooled and they were homogenized using a glass pressure homogenizer. The cell lysates were filtered with a 40-μm cell strainer (Jet Biofil, China). Then, the lysate was passed through a G24 needle 5 times. The equal amount of 100% Percoll was added and centrifuged at 1,500 xg for 15 min at room temperature (RT). Mature spores of EHP were separated from other developmental stages by ultracentrifugation at 12,656 xg, RT, for 15 min in a discontinuous density gradient, including 25%, 50%, 75%, and 100% Percoll (from top to bottom) (Cytiva, USA). The EHP mature spores were kept at 4°C before use.

## Total RNA isolation and cDNA synthesis

Total RNA was extracted from shrimp hemocytes using a FavorPrep Blood/Cultured Cell Total RNA Mini Kit (Favorgen, Taiwan). Then, the first-strand cDNA was synthesized using a First-strand cDNA Synthesis Kit (Thermo Fisher Scientific, USA) according to the manufacturer's instructions. cDNA was stored at −20˚C until use.

## Expression analysis of *Litopenaeus vannamei Toll-like receptors* (*LvTLRs*) and *LvDOME* related genes after EHP challenge

Specific pathogen-free (SPF) *L. vannamei* were separated into control and EHP-infected groups. EHP-infected shrimps were prepared by co-habitation technique [9]. Fifty SPF shrimps were reared together with 15–20 EHP-positive shrimps. Shrimp hemocytes from each group (n = 5) were collected at 1, 9, 11, and 15 days post cohabitation. Total RNA from the hemocytes was extracted and cDNA was synthesized as described above. The amount of cDNA used in each qPCR reaction was 10 ng. The expressions of *LvTLRs* and *LvDOME* signaling pathways, including *LvTLR2* (JN180637.1), *LvMyD88* (JX073567.1), *LvDorsal* (FJ998202.1), *LvDOME* (KC346866.1), *LvJAK* (KP310054.1), *LvSTAT* (KC779541.1), and antimicrobial peptide (AMP) genes, including *LvALF1* (MF135540.1), *LvPEN3* (DQ206403.1), and *LvLyz-c* (AY170126.2) were investigated using a Luna Universal qPCR Master Mix (New England Biolabs, UK) and a CFX96 TouchÔ Real-time PCR Detection System (Bio-Rad, USA) with qPCR specific primers (S1 Table). Relative expression was calculated using the 2−ΔΔCt method relative to the elongation factor 1α (EF1α) gene. The ΔΔCt value was calculated as CtEHP-infected group − CtSPF group.

## Double stranded RNA (dsRNA) production and RNA interference (RNAi) experiment

To prepare specific dsRNAs for the gene knockdown experiment, fragments of *LvTLR2* and *LvDOME* were synthesized by T7 RiboMAX Express RNAi System kit (Promega, USA) using gene-specific primers flanked with the T7 promoter sequence (S1 Table). *GFP* was used as a control. The quality of dsRNA was verified after annealing by agarose gel electrophoresis and the concentration was measured using a NanoDrop 2000 spectrophotometer (Thermo Scientific, USA).

To test the silencing efficiency of dsRNAs, each shrimp was double injected with ds*LvTLR2* (2.5 µg/g shrimp), ds*LvDOME* (1 µg/g shrimp), or ds*GFP* (2.5 µg/g shrimp) in the third abdominal segment. The second injection was carried out 24 hours after the first injection. Then, the injected shrimps were reared together with 15–20 EHP-infected shrimps. Hemolymph was collected at 1, 7, 9, and 11 days after dsRNA double injection. Then, hemolymph was centrifuged at 800 xg for 10 min at 4˚C to collect hemocyte cells. Shrimp hemocyte was collected for total RNA extraction and first-strand cDNA synthesis as described above. The amount of cDNA used in each qPCR reaction was 10 ng. The cDNA samples were used to investigate the *LvTLR2* or *LvDOME* suppression efficiency via RT-qPCR using specific forward and reverse primers (S1 Table). The gene expression levels of ds*LvTLR2*- or ds*LvDOME*-infected shrimps were calculated in relative to that of the ds*GFP*-injected group.

## Detection of EHP copy numbers by absolute qPCR

To determine the copy number of EHP in the infected shrimp samples, the genomic DNA was isolated from the hepatopancreas using a FavorPrep Tissue Genomic DNA Extraction Mini Kit (Favorgen, Taiwan). Fifteen nanograms of the gDNA from five individual shrimp from

each time point were used as templates in the qPCR analysis using a Luna Universal qPCR Master Mix (NEB, UK). The instructions from the company were followed. Small subunit of ribosomal RNA (SSU rRNA) gene (FJ496359.1) was used for generating a standard curve. The copy numbers of EHP in each sample were calculated using the Ct values of each sample based on the SSU rRNA standard curve.

## Cell culture and luciferase reporter assay

To confirm that the *LvLyz-c* production is regulated under the JAK/STAT signaling pathways, a luciferase reporter assay was performed. Human embryonic kidney 293 (HEK293) cells were cultured in a Dulbecco's Modified Eagle's medium (DMEM) (Life Technologies, USA) with 10% heat-inactivated fetal bovine serum (FBS) (Life Technologies, USA). Cells were incubated at 37˚C with 5% $CO_2$ supplement. The cDNA sequence coding for a *LvSTAT* gene was cloned into pcDNA3.1-Myc expression plasmids (Santa Cruz Biotechnology, USA). The pGL3-*LvLyz-c* was constructed by cloning a fragment of the *LvLyz-c* promoter region (position −213 to +24 bp) into a pGL3 luciferase reporter plasmid. For reporter assays, HEK293 cells ($2 \times 10^5$ cells/well) were seeded into 24-well plates. After 24 h, cells were co-transfected with 100 ng of pGL3-*LvLyz-c*-WT (with wild-type *STAT* binding sites) or pGL3-*LvLyz-c*-ΔSDS (containing corresponding deletion of *STAT* binding sites) luciferase reporter plasmids and 1 μg of pcDNA3.1-*Myc-LvSTAT* protein expression plasmids using a Lipofectamine 3000 in Opti-MEM with a 1:1 ratio. Ten nanograms of pRL-TK Renilla luciferase reporter plasmid was transfected and used as an internal control. After 24 h post transfection, the activities of the firefly and renilla luciferases were measured according to the user instruction. Luciferase activities were measured using a TriStar2 LB 942 Multi-mode microplate reader (Berthold). Note that the consensus recognition motif for STAT (TTCNNNGAA) was obtained from Mitchell et. al 2005 [40].

## Recombinant protein production and purification

To investigate the role of *Lv*Lyz-c against EHP infection, recombinant *Lv*Lyz-c (r*Lv*Lyz-c) (AY170126.2) was produced in *E. coli* Rosetta(DE3)pLysS. The protein coding domain of *Lv*Lyz-c was analyzed using a Simple Modular Architecture Research Tool [52]. The nucleotide sequence encoding for a lysozyme-specific domain was amplified using gene-specific primers containing *Nco*I and *Xho*I restriction sites (S1 Table). In addition, a 6x His-tag was added at N-terminus. The r*Lv*Lyz-c fragment was ligated into a pET19b vector (Novagen, USA) and transformed into *E. coli* Rosetta(DE3)pLysS. The recombinant r*Lv*Lyz-c expression was induced with 1 mM of isopropyl-d-1-thiogalactopyranoside (IPTG) for 4 hours at 37˚C. The cells were collected, suspended in 8M urea dissolved in 1X STE buffer (10 mM Tris, 1 mM EDTA, 100 mM NaCl, pH 8.8), and lysed by ultrasonication method. The protein was then purified using a Ni-NTA affinity column under denaturation conditions. The r*Lv*Lyz-c was eluted with 300 mM imidazole in 8M urea. The urea concentration was decreased by using an Amicon Ultra-15 centrifugal filter with a molecular weight cutoff of 3 kDa (Millipore, USA). The purity of the r*Lv*Lyz-c was analyzed on 15% (w/v) SDS-PAGE and western blot using an anti-His tag antibody.

## Antimicrobial assay

Antimicrobial activity of r*Lv*Lyz-c was assessed using an agar well diffusion method as described by Valgas et al, 2007 [53]. The bacterial density was adjusted to an $OD_{600}$ of ~0.2 with the STE buffer and the culture was spread onto the surface of the 2% agar plate. Ten micrograms of r*Lv*Lyz-c were added into a 0.4-cm well and incubated at 30˚C for 16 hours.

The diameters of the lysis clear zone were measured using ImageJ software. Kanamycin, amphotericin b, and hen egg-white lysozyme (HEWL) were used as positive controls, while STE buffer alone was used as a negative control.

## Effect of r*Lv*Lyz-c on EHP spores

To investigate the chitinolytic activity of r*Lv*Lyz-c on the EHP spores, 5 μl of EHP mature spores ($10^8$ spores/ml) were incubated with 250 μl of r*Lv*Lyz-c in different concentrations, namely 0.1, 0.4, 1, 4, 8, and 16 μM for 16 hours at 30°C. The EHP spores were collected by centrifuging at 1,500 xg for 10 min at room temperature, and the supernatant was removed. The spores were resuspended in 1 ml of 1X phosphate buffered saline (PBS) pH 7.4 and stained with 0.5 μl of calcofluor white (1 g/L) for 15 min at room temperature. The spores were collected by centrifuging at 1,500 xg for 10 min at room temperature and washed with 1X PBS twice. 5 μl of the stained spores was gently placed onto a glass slide, sealed with a coverslip, and then observed under a fluorescence microscope (Zeiss Axio Observer 7, Germany) with a 100 × oil objective lens. The excitation wavelength of 385 nm with 20% laser intensity and the exposure time of 20 ms were used for image acquisition.

The chitinolytic activity of r*Lv*Lyz-c was further investigated by a transmission electron microscopy (TEM). The TEM samples were prepared by the Scientific and Technological Research Equipment Centre, Chulalongkorn University. In brief, r*Lv*Lyz-c treated-spores were fixed in a sodium cacodylate buffer (pH 7.2) containing 2.5% glutaraldehyde and 2% paraformaldehyde. Then, fixed spores were post-fixed with 1% osmium tetroxide ($OsO_4$) and embedded in a 2% melted agar. After dehydration using a gradient of cold ethanol. The samples were transferred into an epoxy resin then stained with uranyl acetate and lead citrate to increase the contrast. Spore sections were observed under a transmission electron microscope (Hitachi HT7700, Japan) and imaged with a nominal magnification of 22,000x.

## Effect of r*Lv*Lyz-c on EHP polar tube germination

Five μl of EHP mature spores ($10^8$ spores/ml) was incubated with 250 μl of r*Lv*Lyz-c in different concentrations, including 0, 0.3125, 0.625, 1.25, 2.5, and 5 μM for 16 hours at 30°C. The spores were collected by centrifuging at 1,500 xg for 10 min at room temperature, and the supernatant was discarded. Then, spore germination was induced by the addition of 10 μl of 2% (w/v) Phloxine B (Sigma, USA). Spores were then placed onto a glass slide, sealed with a coverslip, and observed under a Zeiss Axio Observer 7 with a 100 × oil objective lens, and exposure time of 1 ms.

## Image analysis

Image J software [54] was used to measure an area, integrated density and the mean fluorescence of the EHP stained spores for each condition. The total corrected cellular fluorescence (TCCF) was calculated by subtracting the area of selected cell × mean fluorescence of background readings from the integrated density values. Spore wall thickness was measured using a "straight line" tool. A line between electron-dense exospore layer and electron-lucent endospore layer was made and measured. Spore wall area was measured using a "polygon selections" tool. The length was reported in a μm unit. Graphs were plotted using a GraphPad Prism 9 software.

## Statistical analyses

The GraphPad Prism 9 software was used for all statistical analyses. In the knockdown and polar tube germination experiments, a two-tailed unpaired Student's t-test was used to

compare the differences between two groups. For fluorescence intensity analyses, one-way ANOVA was used to analyze the differences compared with the control.

## Supporting information

**S1 Fig. The expression and purification of recombinant *Lv*Lyz-c.** The r*Lv*Lyz-c was analyzed on 15% SDS-PAGE and the western blot was performed using an anti-his antibody. r*Lv*Lyz-c was expressed from the *E. coli* stain Rosetta(DE3)pLysS transformant with (+) and without (−) IPTG induction. Lanes (P) are purified proteins. Lanes M are standard protein size markers. (DOCX)

**S1 Table. Primers used in this study.**
(DOCX)

**S1 Data. Source data for graphs in this study.**
(XLSX)

## Acknowledgments

The authors thank the Marine Shrimp Broodstock Research Center II (MSBRC-2), Charoen Pokphand Foods PCL, for providing the experimental animals.

## Author Contributions

**Conceptualization:** Nutthapon Sangklai, Premruethai Supungul, Pattana Jaroenlak, Anchalee Tassanakajon.

**Formal analysis:** Nutthapon Sangklai, Pattana Jaroenlak, Anchalee Tassanakajon.

**Funding acquisition:** Pattana Jaroenlak, Anchalee Tassanakajon.

**Investigation:** Nutthapon Sangklai.

**Methodology:** Nutthapon Sangklai, Pattana Jaroenlak, Anchalee Tassanakajon.

**Resources:** Premruethai Supungul.

**Supervision:** Pattana Jaroenlak, Anchalee Tassanakajon.

**Writing – original draft:** Nutthapon Sangklai, Pattana Jaroenlak, Anchalee Tassanakajon.

**Writing – review & editing:** Nutthapon Sangklai, Pattana Jaroenlak, Anchalee Tassanakajon.

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
