## [Decision Letter · Decision Letter 0]

13 Mar 2024

Dear Dr. Jaroenlak,

Thank you very much for submitting your manuscript "Immune signaling of Litopenaeus vannamei c-type lysozyme and its role during microsporidian Enterocytozoon hepatopenaei (EHP) infection" for consideration at PLOS Pathogens. As with all papers reviewed by the journal, your manuscript was reviewed by members of the editorial board and by several independent reviewers. In light of the reviews (below this email), we would like to invite the resubmission of a significantly-revised version that takes into account the reviewers' comments.

The manuscript has received two thoughtful reviews. They raise several technical concerns that will need to be addressed in the revision. It is likely that some of these may require some additional experiments or the repetition of work presented with additional controls.

We cannot make any decision about publication until we have seen the revised manuscript and your response to the reviewers' comments. Your revised manuscript is also likely to be sent to reviewers for further evaluation.

Sincerely,

Francis Michael Jiggins

Academic Editor

PLOS Pathogens

James Collins III

Section Editor

PLOS Pathogens

Michael Malim

Editor-in-Chief

PLOS Pathogens

orcid.org/0000-0002-7699-2064

The manuscript has received two thoughtful reviews. They raise a few technical concerns that will need to be addressed in the revision. It is possible that some of these may require some additional experiments or the repetition of work presented with additional controls.

Reviewer's Responses to Questions

**Part I - Summary**

Reviewer #1: This paper seeks to provide insight into the role of shrimp lysozyme-c type (LvLyz-c) in the immune response to infection by the microsporidian pathogen Enterocytozoon hepatopenaei (EHP). Although LvLyz-c was upregulated upon EHP infection, the authors investigated the expression of immune signaling pathways that could regulate LvLyz-c expression during infection. They conclude that the shrimp Domeless promotes LvLyz-c expression and inhibits EHP proliferation. By performing luciferase assays, the authors conclude that LvLyz-c is regulated by the LvSTAT, supporting their claim that JAK/STAT signaling is required for LvLyz-c expression during EHP infection. By heterologous expression of LvLyz-c in bacteria, the authors determine that LvLyz-c has antibacterial and anti-fungal properties. In addition, using the recombinantly expressed LvLyz-c, the authors conclude that LvLyz-c destroys the chitin-containing endospore of EHP and reduces polar tube firing.

As EHP is a significant problem for commercial shrimp farms, it is commendable that the authors investigate the immune response of shrimp to EHP to identify host targets for infection intervention. However, the authors should address many major concerns (detailed below).

Reviewer #2: Sangklai et al. provide an intriguing study that suggests a chitinase activity of invertebrate Lysozyme from shrimp. In reviewing this article, I've come to learn that anti-Candida effects of lysozyme are well-described using vertebrate lysozyme, but less description of antifungal activity has been given for invertebrate lyszyome families (to my knowledge). This study therefore provides a valuable contribution to the literature on invertebrate lysozyme. The authors have also done a good job to present a concise and convincing series of experiments that nicely characterize the anti-Candida effects of LvLyz-c, including an investigation of precisely how it affects Candida endo- and exospore layers.

Overall, the study is convincing and sticks to presenting the core observations of the question at hand. I have only minor comments. A couple of my suggestions arise from my own work, so I'll forego anonymity here and note my identity as Mark A. Hanson.

**Part II – Major Issues: Key Experiments Required for Acceptance**

Reviewer #1: 1) With the cohabitation setup, having a standardized amount of EHP spores used across infections seems impossible. This is a major concern when interpreting any data generated from infected shrimp.

2) For all qRT-PCR results, the authors normalized the gene expression in infected shrimp to uninfected shrimp. Without seeing the raw Ct values for these experiments, it is difficult to conclude that the expression of a given gene changed due to the infection or just time. The authors should identify a gene that does not change expression due to infection or time and normalize to that for both infected and uninfected samples. It seems they did include EF1� as a control, so it is unclear why it was not used for normalization. In addition, the amount of cDNA used per reaction should be included in the materials and methods.

3) It is unclear why the authors did not quantify EHP expression when RNAi against LvLyz-C is performed. The expression of multiple genes will decrease by RNAi against LvDome, including Lv-Lyz-c, and the authors cannot conclude that Lv-Lyz-c inhibits infection without knocking it down.

4) The luciferase assay of LvSTAT and Lv-Lyz-c promoter demonstrates a 2.88-fold increase in activity, which is relatively low. The authors should quantify luciferase activity when only the pGLE3 plasmid is present to eliminate the possibility that LvSTAT is not required for the slight increase in activity reported. In addition, it is unclear why the authors did not normalize to Renilla for each condition in the assay.

5) To quantify the endospore layer size, the authors draw a line between the exospore and endospore and quantify its length. A more accurate measurement would be to calculate the area of the endospore in treated vs untreated spores.

Reviewer #2: (No Response)

**Part III – Minor Issues: Editorial and Data Presentation Modifications**

Reviewer #1: (No Response)

Reviewer #2: -------------------------------

Minor comments:

Line 148: “This highlighted that LvLyz-c is one of the important AMPs to fight against the EHP infection.”

I would discourage this line of thinking. My own work has shown that AMPs are co-regulated due to a need for concise gene regulation, but gene expression should not be inferred as evidence of importance or even of relevance.

That’s not to say induced genes can’t be important, of course they can. The authors characterisation of LvLyz-c is also convincing. But there are tens to hundreds of induced genes after a given infection like EHP, and many of those genes likely contribute little, if anything, to host resistance/tolerance to EHP. I recently wrote a review describing this point in detail reflecting on studies by myself and others using Drosophila, nematodes, and more, available as a preprint and soon to be published in Philosophical Transactions of the Royal Society B. See: https://ecoevorxiv.org/repository/view/6138/

The authors could simply reword their sentence to avoid implying induction is evidence of effect. I really want to emphasize that I don't want to coerce a citation here. I just ask the authors to avoid this logical leap that induction is somehow evidence of function against the inducer.

Fig. 1: it’s striking how the temporal patterns of LvALF1, LvLyz-c, LvDOME, LvSTAT etc… have such tight bounds to their variance, but 11 days post-cohabitation seems to be much lower than both 9 and 15 days post-cohabitation.

First a technical question: can I just confirm how the samples were collected? What is their level of study design independence?

Second: if the authors argue these are truly independent samplings, then could I ask the authors to explain how they imagine such up-and-down induction of immunity might be driven?

Line 183: could the authors indicate where this STAT binding site motif comes from with a reference? Also as a comment on Figure 3: very convincing results! Well done.

Some suggested references that may be relevant to discussion in this study:

Sowa-Jasiłek et al: https://www.sciencedirect.com/science/article/pii/S0944501316300799?via%3Dihub

- A dedicated investigation of G. mellonella lysozyme activity against C. albicans

Marra et al: https://journals.asm.org/doi/full/10.1128/mbio.00824-21?rfr_dat=cr_pub++0pubmed&url_ver=Z39.88-2003&rfr_id=ori%3Arid%3Acrossref.org

- A genetic deletion study in Drosophila that compares the effects of Lysozymes or AMPs on microbiome structure, finding AMPs control G- bacteria while Lysozymes control G+ bacteria

A couple additional references the authors could consider. Do the authors’ results provide insight into what is going on in these shrimp infection systems:

- Shrimp PGN: https://journals.plos.org/plospathogens/article?id=10.1371/journal.ppat.1010967

- anti-LPS factorD -> antiviral?: https://link.springer.com/article/10.1007/s42995-021-00113-y

PLOS authors have the option to publish the peer review history of their article (what does this mean?). If published, this will include your full peer review and any attached files.

Reviewer #1: No

Reviewer #2: **Yes: **Mark A. Hanson
---

## [Editor Report · Decision Letter 1]

16 Apr 2024

Dear Dr. Jaroenlak,

We are pleased to inform you that your manuscript 'Immune signaling of Litopenaeus vannamei c-type lysozyme and its role during microsporidian Enterocytozoon hepatopenaei (EHP) infection' has been provisionally accepted for publication in PLOS Pathogens.

Best regards,

Francis Michael Jiggins

Academic Editor

PLOS Pathogens

James Collins III

Section Editor

PLOS Pathogens

Michael Malim

Editor-in-Chief

PLOS Pathogens

orcid.org/0000-0002-7699-2064
---

## [Editor Report · Acceptance letter]

23 Apr 2024

Dear Dr. Jaroenlak,

We are delighted to inform you that your manuscript, "Immune signaling of Litopenaeus vannamei c-type lysozyme and its role during microsporidian Enterocytozoon hepatopenaei (EHP) infection," has been formally accepted for publication in PLOS Pathogens.

Best regards,

Michael Malim

Editor-in-Chief

PLOS Pathogens

orcid.org/0000-0002-7699-2064